# Prognostic value of preoperative circulating tumor cells counts in patients with UICC stage I-IV colorectal cancer

Thaer S. A. Abdalla[1]*, Jan Meiners[1], Sabine Riethdorf[2], Alexandra König[3], Nathaniel Melling[1], Tobias Gorges[2], Karl-F. Karstens[1], Jakob R. Izbicki[1], Klaus Pantel[2], Matthias Reeh[1]

1 Department of General, Visceral and Thoracic Surgery, University Medical Center Hamburg-Eppendorf, Hamburg, Germany, 2 Department of Tumor Biology, University Cancer Center Hamburg, Center for Experimental Medicine, University Medical Center Hamburg-Eppendorf, Hamburg, Germany, 3 Department of General Surgery, Hospital Wilhelmshaven, Wilhelmshaven, Germany

* t.abdalla@uke.de

**Data Availability Statement:** All relevant data are within the manuscript and its Supporting Information files.

## Abstract

Colorectal cancer (CRC) is one of the leading causes of cancer death worldwide. There is an urgent need to identify prognostic markers for patients undergoing curative resection of CRC. The detection of circulating tumor cells in peripheral blood is a promising approach to identify high-risk patients with disseminated disease in colorectal cancer. This study aims to evaluate the prognostic relevance of preoperative CTCs using the Cellsearch® system (CS) in patients, who underwent resection with curative intent of different stages (UICC I-IV) of colorectal cancer. Out of 91 Patients who underwent colorectal resection, 68 patients were included in this study. CTC analysis was performed in patients with CRC UICC stages I-IV immediately before surgery. Data were correlated with clinicopathological parameters and patient outcomes. One or more CTCs/7.5 mL were detected in 45.6% (31/68) of patients. CTCs were detected in all stages of the Union of International Cancer Control (UICC), in stage I (1/4, 25%), in stage II (4/12, 33.3%), in stage III (5/19, 26.3%) and in stage IV (21/33, 63.6%). The detection of $\geq$ 1 CTCs/ 7.5ml correlated to the presence of distant overt metastases (p = 0.014) as well as with shorter progression-free (*p* = 0.008) and overall survival (*p* = 0.008). Multivariate analyses showed that the detection of $\geq$ 1 CTCs/ 7.5ml is an independent prognostic indicator for overall survival (HR, 3.14; 95% CI, 1.18–8.32; *p* = 0.021). The detection of CTCs is an independent and strong prognostic factor in CRC, which might improve the identification of high-risk patients in future clinical trials.

## Introduction

Colorectal cancer (CRC) is the 3rd most diagnosed cancer and the 4th leading cause of death in the world with 1.3 million new cases annually [1]. Two-thirds of the patients with CRC present with localized and potentially curable disease at diagnosis (TNM Stage I-III) [2]. In these patients, surgery remains the most important treatment modality.

Currently, the overall 5-year survival is 65% [1, 2]. This increased over the last four decades after the introduction of screening programs and the concept of adjuvant chemotherapy [3, 4].

**Funding:** The author(s) received no specific funding for this work.

**Competing interests:** The authors have declared that no competing interests exist.

The surgical technique also evolved with the introduction of total mesorectal excision (TME) for rectal cancer and central venous ligature (CVL) with complete mesocolic excision (CME) for colon cancer, where sharp dissection in the embryological planes increases lymph node yields, subsequently improving staging and survival [5, 6]. Nevertheless, tumor recurrence or spread to distant sites and formation of metastases still occur in 20% of the patients and is the leading cause of death in these patients [7]. Even in patients with apparently early stages TNM (I-II), local recurrence or distant metastases occur despite proper treatment [8].

Tumor persistence and progression occur mainly due to circulating tumor cells (CTCs) which are seeded from the primary tumor and target distant organs, where they eventually mature and cause secondary metastasis [9]. Preoperative identification of CTCs through liquid biopsy is of proven utility in predicting prognosis in breast, colon, and prostate cancer [10–12].

The 8th AJCC Cancer Staging Manual expanded the definitions of Tis, T4a, and M1 and nodal micrometastasis in CRC [13]. However, unlike in breast cancer, new staging categories like M0 (i+), in which CTC or disseminated tumor cells in the bone marrow are detected, are still lacking [14].

Identifying CTCs preoperatively in liquid biopsies could provide information to solve common dilemmas in CRC [15], like patient selection for adjuvant chemotherapy in stage II CRC after tumor resection.

Isolation and molecular analysis of CTCs in peripheral blood are promising approaches to identify disseminated disease, to upgrade or downgrade the multimodal therapy. The Cell-Search® (CS) (Menarini, Silicon Biosystems, Bologna, Italy), a known method for quantification of CTCs based on the expression of the epithelial cell adhesion molecule (EpCAM) and of keratin, is the first standardized system approved by the U.S. Food and Drug Administration for capturing and detection of CTCs derived from metastatic breast and prostate cancer as well as metastatic CRC [10, 16].

The aim of this study is to assess the preoperative value of CTCs in different stages of CRC on the overall survival and progression-free survival using CS.

## Materials and methods

This prospective study was conducted at the University Hospital Hamburg-Eppendorf in Germany. The study was approved by the medical ethics committee of the Chamber of Physicians of Hamburg. All patients gave their written informed consent for inclusion before they participated in the study. The study was conducted in accordance with the Declaration of Helsinki.

The inclusion criteria were: 1) adult patients ($\geq$ 18 years) and 2) presence of a primary resectable CRC. Patients with synchronous malignancy were excluded. Also, patients with resectable metachronous metastasis of CRC or local recurrence were excluded (Fig 1). Out of the 91 patients who underwent colorectal resection, 68 were eligible for inclusion in this study.

Peripheral blood samples for CTC analysis were collected immediately before surgery and within the first 96 hours postoperatively. Follow up was conducted according to S3 German Guidelines [17]. Events considered were death, local recurrence, and distant metastasis. Overall survival was the time from operation to death or last follow-up, and progression-free survival was defined as the time from operation to the diagnosis of tumor recurrence.

### CTC analysis

CTC analysis was performed using CS as previously described [18]. Blood samples (7.5 mL) were collected in CellSave preservative tubes, stored at room temperature, and processed within 96 hours after blood collection, according to the manufacturer's instructions. The

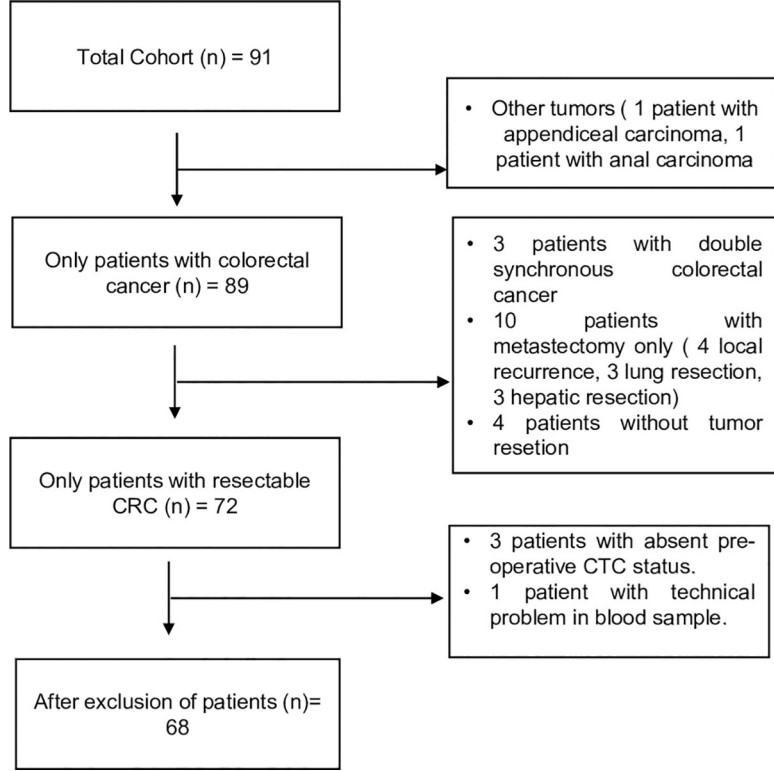

**Fig 1. Flow chart.** Process of patient selection.

accuracy and reproducibility of CS have been described previously [18, 19]. The presence of a nucleus, cytokeratin expression, round or oval cell morphology of cells with a diametre of at lest 4 μm, and absence CD45 expression served as the criteria for CTCs [18].

## Statistical analysis

SPSS statistic software version 25 was used. Histological characteristics were expressed as descriptive statistics. The $x^2$ was used to investigate the association between CTCs and histo-pathological parameters. Survival rates were determined using the Kaplan–Meier method and were compared using the log-rank test. A multivariate analysis of factors that might influence OS was performed using the Cox proportional hazards regression model. The results were presented as hazard ratios with 95% CI. All comparisons were two-tailed. A *p*-value of less than 0.05 was considered statistically significant. All authors had access to the study data and had reviewed and approved the final manuscript.

## Results

### Patient characteristics and CTC detection

CTC analysis was performed in blood samples from 68 patients preoperatively. 38 patients had CTC analysis within the first 96 hours after the procedure. Forty-three patients were nodal positive. Distant metastasis was present in 34 patients at the time of operation (Table 1).

The median age of patients was 64.7 years (range, 18–88 years). Applying a cut-off of $\geq 1$ CTC, 31 out of 68 (45%) patients were CTC-positive preoperatively and 13 out of 38 (19.1%)

**Table 1. Site of metastasis.**

| Site of Metastasis | Number of Patients |
|---|---|
| Lungs | 4 |
| Liver | 21 |
| Peritoneum | 9 |

patients were CTC-positive postoperatively. In 8 out of 13 patients, CTCs were also present preoperatively.

We assessed the correlation between CTC detection preoperatively with sex, age and the following histopathologic parameters: Grade of differentiation (G), tumor Invasion (T), nodal status (N), metastases (M), Union of International cancer control (UICC-Stage), HER2, KRAS, MSI mutations and tumor location (Table 2). The detection of CTCs was related to the

**Table 2. Patient characteristics and correlation of CTCs at baseline with clinicopathological parameters.**

| Variables | Preoperative $\geq$ 1 CTC | | |
|---|---|---|---|
| | **All** | **CTC-Positive** | ***p*-value** |
| All | 68 | 31 (45.6%) | |
| Age | | | 0.812 |
| *<65* | 20 | 8 (40%) | |
| *65–74* | 28 | 13 (46.4%) | |
| *$\geq$75* | 20 | 10 (50%) | |
| Sex | | | 0.320 |
| *Male* | 44 | 18 (40.9%) | |
| *Female* | 24 | 13 (54.2%) | |
| Grade | | | 0.636 |
| *G1* | 1 | 0 (0%) | |
| *G2* | 48 | 23 (49.9%) | |
| *G3* | 13 | 6 (46.2%) | |
| *No Grading** | 6 | | |
| Tumor size | | | 0.270 |
| *T1* | 3 | 0 (0%) | |
| *T2* | 8 | 4 (50%) | |
| *T3* | 38 | 16 (42.1%) | |
| *T4* | 19 | 11 (57.9%) | |
| Nodal status | | | 0.707 |
| *N0* | 25 | 12 (48%) | |
| *N1* | 14 | 5 (35.7%) | |
| *N2* | 29 | 14 (48.3%) | |
| Metastatic stage | | | 0.014 |
| *M0* | 34 | 10 (29.4%) | |
| *M1* | 34 | 21 (61.8%) | |
| UICC Stage | | | 0.065 |
| *Stage I* | 4 | 1 (25%) | |
| *Stage II* | 12 | 4 (33.3%) | |
| *Stage III* | 18 | 5 (27.8%) | |
| *Stage IV* | 34 | 21 (61.8%) | |
| Tumor site | | | 0.291 |
| *Right side* | 22 | 9 (40.9%) | |

(*Continued*)

**Table 2.** (Continued)

| Variables | Preoperative ≥ 1 CTC | | |
|---|---|---|---|
| | **All** | **CTC-Positive** | ***p*-value** |
| *Left side* | 14 | 8 (57.1%) | |
| *Rectum* | 32 | 18 (56.2%) | |
| HER2-Mutation | | | 0.483 |
| *Negative* | 44 | 22(50%) | |
| *Positive* | 4 | 2 (50%) | |
| *Missing* | 20 | 7 (35%) | |
| KRAS-Mutation | | | 0.345 |
| *Negative* | 24 | 10 (41.7%) | |
| *Positive* | 29 | 16(55.2%) | |
| *Missing* | 15 | 5 (33.3%) | |
| MSI | | | 0.718 |
| *Present* | 5 | 2 (40%) | |
| *not present* | 25 | 13 (52%) | |
| *Missing* | 38 | 16 (42.1%) | |

*P*-value Indicates significance according to the χ2 test when CTC-negative patients are compared with CTC-positive patients. Round parentheses indicate percentages. No Grading* after radiotherapy for rectal cancer, instead of Dworak's system of tumor regression

presence of distant metastases *(p = 0.014)* and showed a tendency towards the Union of International cancer control stage *(p = 0.065)*. Other parameters did not significantly correlate with CTC positivity preoperatively. Postoperatively, CTC detection did not correlate with any of the above-mentioned parameters.

## Univariate and multivariate analysis of survival

The median survival time was 32 months. Kaplan-Meier's univariate analysis showed that patients with ≥ 1 CTC had significantly shorter progression-free ($p = 0.008$) as well as overall survival ($p = 0.008$) compared to CTC-negative patients (Fig 2). Also, the presence of distant metastases at baseline was associated with shorter OS (*p*-value 0.002) and shorter PFS (*p*-value < 0.001) (Fig 3). However, postoperative CTCs did not correlate with OS (*p*-value = 0.829) and PFS (*p*-value 0.876) (Fig 4).

Nine clinicopathological factors were analyzed using Cox-Regression analysis. Only 4 factors correlated with survival in univariate analysis (Table 3). These included age, metastatic disease, UICC stage, preoperative CTC detection ($p < 0.05$). Gender, postoperative CTC detection, and other factors were not related to survival in the univariate analysis. The four factors significantly related to survival in univariate analysis were evaluated in the multivariate analysis, which showed that only advanced age and preoperative CTC detection were independent prognosticators of an unfavorable OS (Table 3).

## Discussion

Despite advances in multimodal treatment, CRC is the second most common cause of cancer death worldwide [1]. CTCs play a pivotal role in disease progression, metastasis, and recurrence [20]. Aside from TNM classification and status of resection margin, novel tools and staging systems are needed for adequate prognostic staging and for guiding multimodal therapy

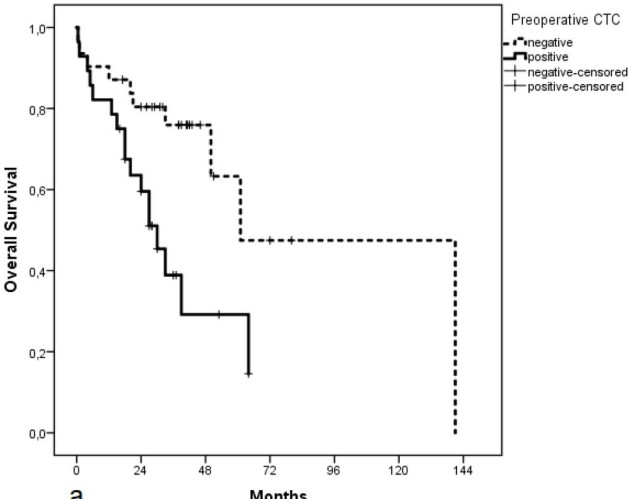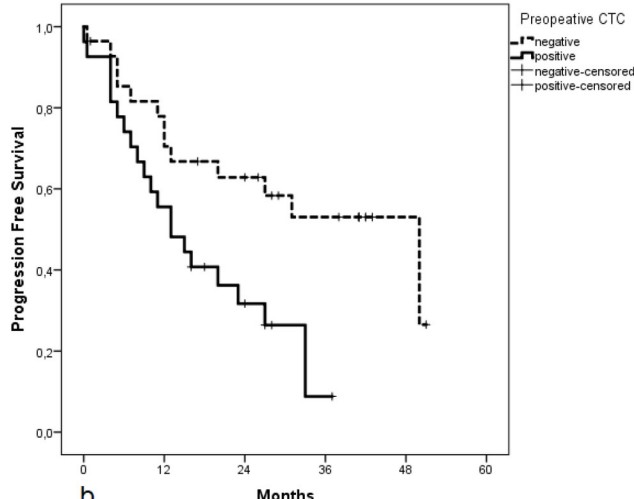

| | Log-Rank (Mantel-Cox) | |
| --- | --- | --- |
| | Chi-Square | *p*-value |
| Figure A (OS) | 6.962 | 0.008 |
| Figure B (PFS) | 7.072 | 0.008 |

**Fig 2. Kaplan-Meier survival analysis for overall survival and progression-free survival according to the preoperative CTCs.** (a): Overall survival in patients with CRC, positive vs. negative preoperative CTCs. (b): Progression-free survival in patients with CRC, positive vs. negative preoperative CTCs. *P*-value Indicates significance according to Log-Rank (Mantel-Cox) test when CTC-negative patients are compared with CTC-positive patients preoperatively.

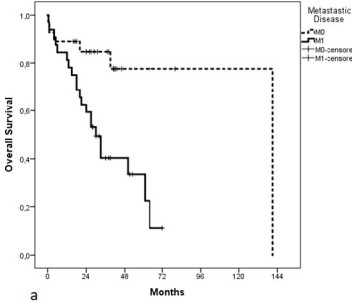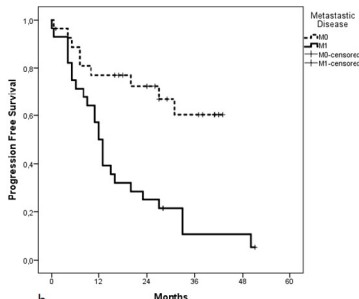

| | Log-Rank (Mantel-Cox) | |
| --- | --- | --- |
| | Chi-Square | *p*-value |
| Figure a (OS) | 9.262 | 0.002 |
| Figure b (PFS) | 12.447 | < 0.001 |

**Fig 3. Kaplan-Meier analysis for overall survival and progression-free survival according to metastatic status.** Fig (a): Overall survival in patients with CRC, metastatic vs. non-metastatic disease. Fig (b): Progression-free survival in patients with CRC, metastatic vs. non-metastatic disease. *P*-value Indicates significance according to Log-Rank (Mantel-Cox) test when mCRC is compared to non-mCRC.

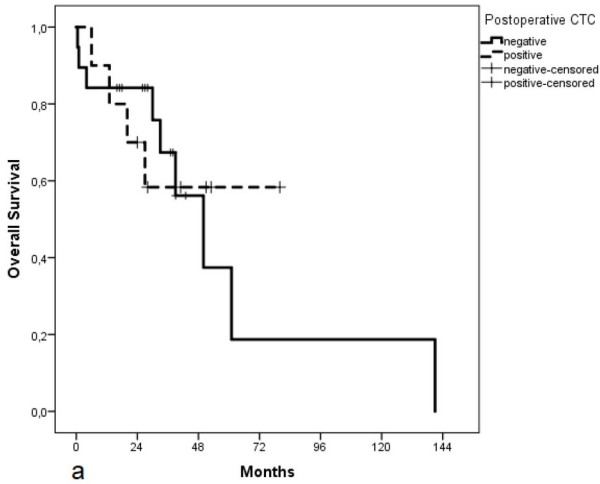
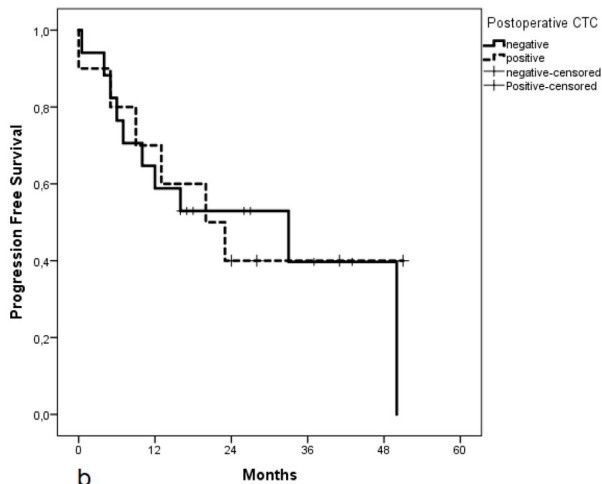

| | Log-Rank (Mantel-Cox) | |
|---|---|---|
| | Chi-square | *p*-value |
| Figure a (OS) | 0.047 | 0.829 |
| Figure b (PFS) | 0.024 | 0.876 |

**Fig 4. Kaplan-Meier analysis for overall survival and progression-free survival according to the postoperative CTCs.** Fig (a): Overall survival in patients with CRC, positive vs. negative postoperative CTCs. Fig (b): Progression-free survival in patients with CRC, positive vs. negative postoperative CTCs. *P*-value Indicates significance according to Log-Rank (Mantel-Cox) test when CTC-negative patients are compared with CTC-positive patients postoperatively.

[21]. CTC identification has proven to be an important prognosticator in breast cancer [22]. For this reason, a decade ago, the 7th AJCC Cancer Staging Manual introduced the new category M0(i) for breast cancer, which is defined by the presence of circulating or disseminated tumor cells not exceeding 0.2 mm detectable in bone marrow, circulating blood, or other non-regional tissues of non-metastatic patients. In contrast, such a category is still lacking in the current CRC Staging system [23], this is because the assessment of CTCs in CRC is still

**Table 3. Univariate and multivariate analysis of overall survival in patients with CRC.**

| | Univariate Analysis | | | Multivariate Analysis | | |
|---|---|---|---|---|---|---|
| | HR | 95% CI | *p* | HR | 95% Cl | *p* |
| Age, <65,65–74, ≥75 | 2.95 | 1.50–5.80 | 0.002 | 2.85 | 1.51–5.38 | 0.001 |
| Sex, male vs female | 0.833 | 0.37–1.87 | 0.650 | 1.12 | 0.49–2.52 | 0.784 |
| Grade of Differentiation, G1-3 | 1.30 | 0.51–3.33 | 0.585 | | | |
| T, T1-T4 | 1.26 | 0.73–2.19 | 0.392 | | | |
| N, N0- N2 | 1.09 | 0.67–1.78 | 0.710 | | | |
| Metastatic stage, M0 vs M1 | 3.01 | 1.11–8.20 | 0.019 | 2.54 | 0.30–21.16 | 0.388 |
| UICC Stage | 2.20 | 1.15–4.18 | 0.016 | 1.04 | 0.33–3.33 | 0.938 |
| Preoperative CTCs, neg vs pos | 2.85 | 1.25–6.46 | 0.012 | 3.14 | 1.18–8.32 | 0.021 |
| Postoperative CTCs, neg vs pos | 1.53 | 0.43–5.51 | 0.515 | | | |

*p* Indicates significance according to cox regression analysis comparing the specified variables. HR indicates hazard ratio.

controversial [16] and different technical platforms have provided conflicting results [24–26]. Here, we applied the FDA-cleared Cellsearch® system, which uses immunomagnetic enrichment (EpCAM) and immunocytochemical (cytokeratin, CD45, DAPI) fluorescence analysis. The epithelial cell adhesion molecule (EpCAM) was first identified in colon cancer in 1979 [27]. It is expressed by a variety of epithelial cells, is highly expressed in colorecal cancer cells [28] and is known to promote tumor expansion and oncogenesis [29]. Thus EpCAM is widely used to capture and detect CTCs. During the process of dissemination and metastasis, tumor cells undergo epithelial-mesenchymal transition (EMT) and might lose their epithelial cell features including EpCAM and/or keratin expression. CTCs that have completely downregulated these epithelial properties are not detectable with the CellSearch system (CS) and require marker-free enrichment methods such as size-, plasticity-, or density-based approaches (Alix-Panabieres C and Pantel K Cancer Discovery 2021) for their identification [30, 31]. Nevertheless, the CellSearch system (CS) which based on EpCAM-related capture of CTCs is standardized, time-efficient and provides clinically relevant results for CTC detection in a wide range of carcinoma patients [31–33]. Several studies have addressed the detectability of CTCs and their prognostic impact in mCRC using CS. According to Cohen et al., CTCs detected using CS provide additional information about tumor burden to imaging studies and therefore have gained a prognostic and predictive value in guiding treatment of mCRC [12]. In addition, Bork et al. showed that CTCs were detectable using CS in patients with CRC (I-III). CTCs were detectable even in patients without nodal or distant metastasis at time of operation (UICC I-II), which was associated with worse survival outcome [34]. In our cohort CTCs were detectable in 45.6% of the patients before surgical resection and were present in all stages of CRC (I-IV). Even in early-stage CRC, UICC I (T1-2, N0, M0), 25% of patients were CTC-positive.

Up to now, there is no consensus regarding the threshold used to define CTC positivity in CRC [16]. Cohen et al have used the cut-off of $\geq$ 3 CTC/7.5 ml for defining CTC positivity in metastatic CRC [12], while others have shown that cut-offs $\geq$1 CTC/7.5 ml and $\geq$ 2 CTC/7.5 ml were also associated with poor prognosis in CRC [35, 36]. We used a strict cut-off of $\geq$1 CTC/7.5 ml. CTC detection before surgery was associated significantly with a shorter progression-free (P = 0.008) and overall survival (P = 0.008). Multivariate analyses identified CTCs as a strong, independent, prognostic indicator for overall survival.

Many studies have shown inferior survival for right-sided tumors [37, 38], which has prompted us to assess the effect of tumor location on CTC detection. Right-sided colon cancer (RCC), left-sided colon cancer (LCC), and rectal cancer (RC) were considered independently [39–41]. Although more CTCs were present in LCC compared to RCC and RC, there was not a statistically significant correlation. A previous publication from Nicolazzo et al. showed similar results [39]. However, larger cohorts are needed to deliver a robust conclusion about tumor sidedness and CTC detection.

From a clinical perspective, using CTCs in peripheral blood allows assessing cancer prognosis in a non-invasive and easily applicable method [16] that allows real-time monitoring of tumor dynamics [42]. Our present study supports the assumption that preoperative CTC detection in CRC identifies patients with shorter OS and PFS, which might contribute to an improved stratification of high-risk CRC patients. Nevertheless, larger validation studies are required before the implementation of CTC detection into tumor staging classification of CRC.

## Conclusion

Preoperative CTC detection is an important prognostic marker for survival in CRC, which can be further developed as an enrichment tool to study a high-risk population of CRC patients in clinical trials.

## Supporting information

**S1 Data. Patients histopathological and survival data.**
(SAV)

## Author Contributions

**Conceptualization:** Klaus Pantel, Matthias Reeh.

**Data curation:** Jan Meiners, Karl-F. Karstens.

**Formal analysis:** Thaer S. A. Abdalla.

**Investigation:** Sabine Riethdorf, Tobias Gorges, Klaus Pantel.

**Supervision:** Alexandra König, Nathaniel Melling, Jakob R. Izbicki, Matthias Reeh.

**Writing – original draft:** Thaer S. A. Abdalla.

**Writing – review & editing:** Sabine Riethdorf, Klaus Pantel, Matthias Reeh.

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
