## [Decision Letter · Decision Letter 0]

22 Apr 2021

PONE-D-21-08416

Prognostic value of preoperative circulating tumor cell counts in patients with UICC Stage I-IV colorectal cancer

PLOS ONE

Dear Dr. Abdalla,

Thank you for submitting your manuscript to PLOS ONE. After careful consideration, we feel that it has merit. Therefore, we invite you to submit a revised version of the manuscript that addresses the points raised during the review process.

In addition to the reviewer's comment, it is strongly recommended to comment and discuss: i) the size and shape of CTCs dectected (ie. presence of clsuters); ii)  the limit of the technique used based on EpCAM marker

We look forward to receiving your revised manuscript.

Kind regards,

Dominique Heymann, Ph.D.

Academic Editor

PLOS ONE

Journal Requirements:

Your ethics statement should only appear in the Methods section of your manuscript. If your ethics statement is written in any section besides the Methods, please delete it from any other section.

Reviewers' comments:

Reviewer's Responses to Questions

**Comments to the Author**

1. Is the manuscript technically sound, and do the data support the conclusions?

Reviewer #1: Yes

2. Has the statistical analysis been performed appropriately and rigorously? 

Reviewer #1: I Don't Know

3. Have the authors made all data underlying the findings in their manuscript fully available?

Reviewer #1: Yes

4. Is the manuscript presented in an intelligible fashion and written in standard English?

Reviewer #1: Yes

5. Review Comments to the Author

Reviewer #1: The authors should state in their discussion that this is not the first study to investigate CTC in non-metastatic colorectal cancer and discuss related work in more detail. E.g. the article "Circulating tumour cells and outcome in non-metastatic colorectal cancer: a prospective study. Br J Cancer. 2015 Apr 14;112(8):1306-13. doi: 10.1038/bjc.2015.88. " should be cited and discussed, as this was the first trial to investigate non-metastatic CTC count in CRC with the Cell search system.

6. PLOS authors have the option to publish the peer review history of their article (what does this mean?). If published, this will include your full peer review and any attached files.

Reviewer #1: No

---

## [Author Response · Author response to Decision Letter 0]

20 May 2021

Dear Prof. Heymann, 

Thank you very much for giving us the opportunity to resubmit our manuscript in your highly respected journal.

In the following you will find point by point answers with regard to the editorial comments as well as the comments by the reviewers. Each table was revised in accordance to the journal’s guidelines. All changes are clearly marked in the revised manuscript.

We would greatly appreciate to publish our work in the “PLOS ONE” and ask for a revised review process.

Sincerely yours,

Thaer S. A. Abdalla

Editorial comments:

1. Comment and discuss the size and shape of CTCs dectected (ie. presence of clusters)

Thank you for your kind remarks. Keratin-positive nucleated cells of round or oval shape with a diameter of at least 4 µm and CD45 negativity were considered CTCs. Only in a minority of cases CTC clusters with 2 or 3 CTCs were detected.

2. Comment and discuss the limit of the technique used based on EpCAM marker

As requested, the limits of the anti-EpCAM methods have been discussed in the lines 174-185 pages 9-10.

3. Changes in the reference list

The list of references has been updated in order to answer the requested changes. The references 27-30 and 32-34 have been recently added. 

Reviewer 1:

First of all we would like to give our thanks to the Reviewer 1 on his thoughtful comments and suggestions.

1. The authors should state in their discussion that this is not the first study to investigate CTC in non-metastatic colorectal cancer and discuss related work in more detail. E.g. the article "Circulating tumour cells and outcome in non-metastatic colorectal cancer: a prospective study. Br J Cancer. 2015 Apr 14;112(8):1306-13. doi: 10.1038/bjc.2015.88. " should be cited and discussed, as this was the first trial to investigate non-metastatic CTC count in CRC with the Cell search system.

Thank you for this remark. We absolutely agree that there are previous studies to investigate CTCs in non metastastic colorectal cancer like the one by Bork et al. Thus, we added a revised description on page 10 of the manuscript. 

In our work we used the CellSearch® system for identifying CTCs in all stages of CRC scheduled for resection with curative intent and compared their detection preoperatively and within 96 hours postoperatively to the OS and PFS.

---

## [Editor Report · Decision Letter 1]

25 May 2021

Prognostic value of preoperative circulating tumor cells counts in patients with UICC Stage I-IV colorectal cancer

PONE-D-21-08416R1

Dear Dr. Abdalla,

We’re pleased to inform you that your manuscript has been judged scientifically suitable for publication and will be formally accepted for publication once it meets all outstanding technical requirements.

Kind regards,

Dominique Heymann, Ph.D.

Academic Editor

PLOS ONE
---

## [Editor Report · Acceptance letter]

2 Jun 2021

PONE-D-21-08416R1 

Prognostic value of preoperative circulating tumor cells counts in patients with UICC Stage I-IV colorectal cancer. 

Dear Dr. Abdalla:

I'm pleased to inform you that your manuscript has been deemed suitable for publication in PLOS ONE. Congratulations! Your manuscript is now with our production department. 

Kind regards, 

on behalf of

Pr. Dominique Heymann 

Academic Editor

PLOS ONE